# Why Should Human-Animal Interactions Be Included in Research of Working Equids’ Welfare?

**DOI:** 10.3390/ani9020042

**Published:** 2019-01-30

**Authors:** Daniela Luna, Tamara A. Tadich

**Affiliations:** Departamento de Fomento de la Producción Animal, Facultad de Ciencias Veterinarias y Pecuarias, Universidad de Chile, Santa Rosa 11735, La Pintana, Santiago 8820000, Chile; danluna@veterinaria.uchile.cl

**Keywords:** animal welfare, attitudes, empathy, human-animal interaction, pain perception, working equids

## Abstract

**Simple Summary:**

Appropriate strategies aimed at improving the welfare of working equids should include the assessment of their welfare status, as well as the identification of the human attributes that influence owner-equine interactions. From a human´s point of view, empathy, attitudes towards animals, perception of animal pain and the owner´s locus of control are some of the psychological attributes that modulate the human-equine relationships that can affect equids’ welfare. There is, however, still little research dedicated to identifying and assessing the owner’s psychological attributes that underlie their behaviours, and that may be implicated in the welfare of their working equids. This review aims to point out why the inclusion of human psychological attributes that modulate human-animal interactions, can benefit welfare research in working equids. We recommend that in order to advance in the improvement of working equids’ welfare on a global scale, an integral understanding of those human psychological attributes that influence the owner’s behaviour and modulate owner-equine interactions must be established and taken into account in future studies.

**Abstract:**

The livelihood of working horses’ owners and their families is intimately linked to the welfare of their equids. A proper understanding of human-animal interactions, as well as the main factors that modulate them, is essential for establishing strategies oriented to improve the welfare of animals and their caretakers. To date, there is still a paucity of research dedicated to the identification and assessment of the human psychological attributes that affect the owner–equine interaction, and how these could affect the welfare of working equids. However, some studies have shown that empathy, attitudes towards animals, human perception of animal pain and the owner´s locus of control are some of the psychological attributes that participate in human-equine interactions and that these can result in poor welfare of working equids. A better understanding of the relationship between human attributes and equids’ welfare can provide an opportunity to improve the quality of interactions between owners and their working equids and thus improve their welfare. This review aims to explain why the inclusion of human psychological attributes that modulate the human-animal interactions can benefit welfare research in working equids. The role that empathy, perception of animal pain and locus of control play in the promotion of good welfare in working equids is emphasized.

## 1. Introduction

The role that equines have played in human societies since their domestication, approximately 6000 years ago, has been crucial for the development of nations and cultures [1]. From war to recreation and sports, to agriculture and transport, equines are among the animals that have contributed the most to the economic development of societies, and also to human well-being [2]. However, despite the increasing use of mechanized means and the technological advances that societies have undergone in the last century, there are still many countries where the use of these animals for work persists, either for draught work or for use as pack animals.

According to the Food and Agriculture Organization of the United Nations (FAO), there are over 112 million equids (horses, donkeys and mules) in developing countries [3]. The majority of these are working animals who support approximately 600 million people worldwide [4]. Equids are considered an essential component of the livelihoods of people living in impoverished urban and rural communities [5,6,7]. These people are often ignored by society and excluded from government policies and decisions, as are their equines [8]. These animals are particularly useful in: (a) reducing poverty through direct or indirect income generation; (b) providing access to food and services; (c) increasing food security; (d) constituting one of the main resources of energy in farming [8]; and (e) increasing the resilience capacity of families that depend on them, by being part of their natural, financial, physical and social capital [5,9]. Therefore, an adequate welfare state is an essential prerequisite for the successful performance of these animals and, consequently, for the well-being of their owners and families [4,7,10], becoming an excellent model of the one welfare framework proposed by García-Pinillos et al. [11,12]. The framework recognizes the links between animal welfare, human wellbeing and the environment, aiming towards the promotion of global objectives such as suffering relief, food security and poverty reduction [11,12].

Working equids are often owned by people who belong to the most impoverished socio-economic strata of society [4,13,14], which results in animals having to work for long hours, under adverse environmental conditions, and often without sufficient essential resources (e.g., water and food availability, veterinary treatment, access to a farrier and appropriate equipment) [10,15,16]. Consequently, these animals suffer from multiple welfare problems [13,16,17]. The financial constraints that commonly characterize working equids owners may have a detrimental effect on the welfare of these animals [8,18]. However, recently a study found that socio-economic constraints of working horse owners in Chile did not affect the welfare condition of the equines, suggesting that other attributes of owners’ might be involved, such as their ability to empathize with animals [13].

During the last three decades, there has been considerable research interest in human-animal interactions (HAIs). It has been described that a thorough understanding of human-animal interactions is an essential component of any strategy oriented to improve the well-being of animals and their caretakers or owners [19]. In fact, there are several studies providing information about the implications that the quality of human-animal interactions have on the welfare and productivity of farm animals [20,21,22,23]. For example, Breuer et al. [20] found that fear of commercial dairy cows towards humans was significantly correlated with milk yield and composition at the farm level. In this sense, milk yield, protein and fat were lower at farms where animals showed less approach towards the experimenter in the standard test. In addition, in recent years, increasing attention has been paid to the factors influencing human-animal interactions. Empathy, attitudes towards nonhuman animals and perception of animal pain are some of the human psychological attributes that play an important role in modulating both human-animal relationships and animal welfare [24,25,26,27,28,29,30,31,32,33]. However, these aspects have been extensively addressed in the context of livestock industries [24,25,26,27,28,29,30,31], while very little research has been undertaken to identify the human attributes that modulate the owner-equine interaction and how these could, eventually, affect working equids’ welfare [13,34,35,36].

Increasing understanding of the human attributes that affect human-equine interaction, and how these impact on the equids´ welfare, can be useful for an appropriate design of intervention strategies oriented to improve the quality of the owner–horse relationship and, consequently, the welfare of working equids. The aims of this review are: (a) to describe the main implications of human-animal interactions on animal welfare; (b) to examine some of the psychological factors that modulate human-equine interactions, placing special emphasis on the role that empathy, attitudes towards animals, human perception of animal pain and the owner´s locus of control have on equine welfare; and finally (c) to highlight why identifying and assessing the human psychological attributes that participate in human-animal interactions could be relevant for welfare research in working equids.

## 2. Human-Animal Interactions and Their Main Implications for Animal Welfare 

The study of human-animal interactions (i.e., Anthrozoology) is an interdisciplinary field of growing interest for both the scientific and general public, that is oriented to quantify the bi-directional effects of the human-animal relationship on the health and well-being of both humans and animals [37]. Hosey and Melfi [38] undertook a comprehensive examination of predominant research themes in this area. Reviewing more than 300 articles, they found that most studies in this field were conducted mainly in the context of companion animals, and secondarily in agricultural animals. In relation to literature about HAIs in companion animals, the greatest emphasis was put on animal-assisted interventions and the benefits (physiological and psychological) of pet ownership for people and interactions with them [38,39,40]. By contrast, agricultural animal research was (and has been to date) more focused on the consequences that HAIs have on animal welfare and productivity in the livestock industries [20,21,22,23,38]. 

The term human-animal interaction is often used to refer to many different things and contexts, including but not limited to companion animal ownership, animal-assisted interventions, and simple contact between a human and nonhuman animal [41]. According to Estep and Hetts [42], it can be defined as the degree of relatedness or distance between animal and humans, relationships developed between the stockperson or caretaker and an animal in his/her care that requires a mutual individual recognition [42]. According to Waiblinger et al. [19], it is a dynamic process where previous interactions between the animal and humans form the foundation of a relationship that then exerts a feedback effect on the nature and perception of future interactions. Research in the livestock industry has documented that the quantity and quality or nature of interactions between animals and the people responsible for their care and handling, has a substantial effect on the behaviour, physiology, welfare and productivity of farm animals [20,21,22,23,43,44,45]. Interactions by humans may be neutral, positive or negative in nature [19]. In this context, numerous studies have shown that negative interactions that stimulate fear responses in animals can become a source of stress [22,23,44,45]. This not only hampers the handling of animals, which in turn endangers human caregivers and animals, but also has negative effects on animals´ welfare, increasing disease susceptibility and decreasing productivity [20,21,22,23,44,45]. In contrast, positive human interactions, such as gentle tactile contacts, regular visual contact or food rewards, can significantly reduce animals’ fear of humans. Such practices may enhance the productivity [23,46] and the health of farm animals [47]. On the other hand, human-animal interactions can also be strongly influenced by human attributes, such as the attitudes, personality traits or job satisfaction. These aspects can affect the behaviour of stockpersons towards animals and, in consequence, production and animal welfare. [24,48,49,50]. Therefore, strategies aimed at improving the welfare of animals can benefit from an appropriate understanding of HAIs, particularly if we determine and understand what the underlying factors that modulate these interactions are [19]. Pritchard et al. [15] have shed light on the importance that the assessment of the human-animal interaction has for establishing appropriate strategies to improve the welfare of working equids and their owners. The authors propose that “without a degree of social bonding between the owner or user and his animal (…), there is little motivation to improve welfare”. Similarly, Hemsworth et al. [34] provided a review of the relationship between horse owners’ attributes and the welfare of horses used for recreational purposes. The authors concluded that understanding the relationship between owners’ attributes (e.g., their attitudes towards animals) and horses’ welfare would provide an opportunity to improve the quality of the human-horse relationship and, subsequently, the welfare of these animals [34]. To date, most studies have mainly been limited to identifying the welfare problems and establishing the main risk factors that affect the welfare of working equids [13,15,16,51,52,53,54,55,56,57,58], without the in-depth assessment of the human psychological attributes involved in the owner–equine interaction, and the implications that these could have for the welfare of equids.

### Human-Equine Interactions

Human-equine interactions have been varied through history depending on human needs [59,60]. The interaction between humans and horses can be viewed in a spectrum, from short occasional interactions, such as those between veterinarians and horses, to long-term bonds that occur between an owner and their horse [60]. However, although there has been a growing scientific interest in human-equine interactions, and several studies examining the human–horse relationships are available [61,62,63,64], this field has received less attention compared to the human-livestock interactions [19,21]. In addition, most studies that address the human-equine interaction and equine welfare have been focused on how the animals perceive humans [13,52,53,55,65,66], and several authors have used diverse methods to assess the human-equine relationship from the animal’s perspective. These assessments have commonly included behavioural tests to assess reactions of equines to human approach, touch and handling [15,52,53,55,57,58,65,66,67], as well as physiological measures, such as heart rate [68] and cortisol levels in response to human interactions [69,70]. However, although it has been described that human-equine interactions have the potential to impact upon the welfare of both the animal and its owner [60], the assessment of human-equine interactions from the human perspective (e.g., the owner´s attitudes or empathy towards animals) and their relationship with equine welfare, has not received much attention [34,35]. 

## 3. Factors Implicated in Human-Animal Interactions

Over the past 30 years, since the interactions between humans and nonhuman animals gained scientific interest, several studies have determined the main factors that modulate HAIs. From the animals’ perspective, the previous experiences with humans [71,72], age [73,74], genetics [75,76], and personality traits or temperament [67], are some of the factors that influence HAIs. However, human psychological attributes, such as personality traits [24,77], empathy towards animals and people [25,27,32,77,78,79], attitudes towards animals [24,25,26,27,30,31,32,77,80], human perception of pain in animals [25,28,32,81,82] and locus of control [36] can also influence HAIs and can, consequently, influence animal welfare. Some of the psychological constructs that have been studied in the field of HAI and equine welfare, such as empathy towards animals, human perception of pain in animals, attitudes towards animals and locus of control, will be described. Moreover, we will address how some of these human psychological attributes are implicated in the human–equine interactions and working equids’ welfare. 

### 3.1. Empathy towards Animals 

Within the growing body of HAI research, one particular psychological attribute that is beginning to receive attention is empathy. Empathy can be conceptualized as the ability to understand and share the emotions of others (human or animal), while maintaining an other-self distinction [83,84]. It can be considered as a multidimensional construct with cognitive and emotional aspects [85,86]. From a cognitive point of view, empathy is the ability to recognize and understand what the other is feeling, adopting their perspective, whereas the emotional aspect implies the capacity to experience affective reactions to the observed experiences of other individuals, and share their emotions [83,86]. Therefore, empathy allows interpreting and predicting the behaviours and intentions of others and responding appropriately to them, through altruistic and prosocial behaviours, such as sharing, containing or providing assistance [87,88]. Empathy has also been linked to oxytocin release; moreover, recent findings by Connor et al. [33] suggest, for the first time, an association between allelic variation in the oxytocin receptor gene (*OXTR*) and animal directed empathy in humans. Conversely, the lack of ability to experience feelings of empathy has been linked to antisocial and aggressive behaviours, including animal cruelty [89,90]. 

In contrast to the substantial number of studies on human empathy [80,83,85,86,87,88,89,90,91], human empathy towards animals has received less attention [25,27,29,32,92]. However, research in this field has shown that empathy towards animals plays an important role in modulating both HAIs and animal welfare [25,27,32,35,93]. Research has demonstrated that empathy towards animals is not only associated with the ability to empathize with people [35,79,94], but is also intrinsically related with: (a) positive attitudes towards animals [32,95], (b) greater sensibility with which animal pain is perceived and qualified, both by pet owners [32] and veterinarians [29], and (c) productivity [24] and animal welfare [25,35]. For example, Hanna et al. [24] reported that higher empathy scores were related to higher milk yield in dairy cows. Kielland et al. [25] investigated how farmers’ attitudes and empathy towards animals affected the welfare of dairy cattle. The authors reported that dairy cattle with a lower incidence of skin lesions belonged to farmers who expressed higher levels of empathy towards animal pain. In consequence, the acquisition of empathic abilities of owners towards animals can have important consequences both for the performance and welfare of their animals. Therefore, strategies oriented to improve the welfare of working equids should not only consider the identification of their main welfare problems, but should also include the assessment of the main factors that modulate HAIs from the human perspective, such as the owner’s empathy towards animals. 

In contrast to empathy towards humans, there is still limited information on how empathy towards animals can be measured [29]. At present, three ways to assess and infer the levels of empathy towards animals have been described. These are: (a) the use of self-reported questionnaires or scales, developed from questionnaires used to assess empathy towards humans [27,32,79]; (b) the use of narrative techniques [78,96] or images involving an animal in a situation of need or suffering [25]; and (c) the assessment of physiological responses while viewing movies or images of animals in negative situations or experiencing a painful condition, such as psychophysiological signs (e.g., phasic skin conductance, heart rate, corrugator electromyographic activity) [97], neural responses [98] or changes in hemodynamic activity in brain regions involved in empathic behaviours [99]. 

Recently, a study using a modified version of the Animal Empathy Scale developed by Paul [79] showed that working equines owners’ empathy towards animals seems to play an important role in determining the way animals are treated and cared for in Chile [35]. The authors investigated how the welfare state of working equines, assessed mainly through animal-based indicators, was associated with owners’ levels of empathy. The results of the regression analysis showed that equines in better welfare conditions belonged to owners with higher scores on a human-animal empathy scale [35], in accordance with Kielland et al. [25] findings for dairy cattle. Therefore, despite limited evidence, it is suggested that the level of human-animal empathy might be used as a good predictor of the welfare status of animals. Consequently, the strategies oriented to improve the welfare of working equines should consider the inclusion of education programs aimed at promoting the development of empathic skills in owners and users of working equids, such as children, who commonly participate actively in the husbandry practices associated with their working equids [35,100]. 

The impact of educational programs based on the promotion of empathy and good welfare practices has been described in schoolchildren from different communities in Ethiopia [101,102] and Mexico [100] who use working donkeys, and could also be used for adults. The interventions in primary schools in Ethiopia showed that children tended to increase visits to veterinary hospitals when their animals were sick. They decreased the harmful practices towards their animals such as beating and overloading them [101]. Furthermore, in another similar intervention program, children mentioned that they felt sad when they saw a donkey working with wounds and were more sensitive in responding to these; likewise, they accepted that donkeys feel pain just like humans [102]. Therefore, the evidence shows that empathy has implications for the treatment and care that an animal receives, as well as for the decision making on whether an animal should receive attention when it is in need. In addition, the acquisition of empathic skills in Ethiopian children confirm that empathy allows individuals to relate to the emotional state of another individual, which can lead to prosocial behaviours in order to alleviate the distress of others and promote their welfare [103]. On the other hand, the interventions in schools in the community of Tuliman (Mexico) showed that children, after receiving talks on animal welfare, easily recognized the need to provide donkeys with food and water, but contrary to reports from communities in Ethiopia, rarely recognized the need for veterinary services [100]. Therefore, the inclusion of educational strategies in primary schoolchildren, and during veterinary clinical work in communities, may have a profound effect on how these children and caretakers define their frame of thinking about animals, which could have a positive impact on the welfare of working equids within communities.

### 3.2. Perception of Animal Pain 

Pain is a complex psychological state with a great evolutionary significance; it can be experienced by one individual (human or animal) and perceived in others. In addition, perception of pain in others is considered as a powerful social stimulus capable of triggering an affective state in the individual who perceives the pain, from which empathy can originate [104]. During the last decade, some studies have shown a relationship between empathy towards animals and the scoring of animal pain [29,32,35]. Norring et al. [29] used 13 painful conditions in cattle to investigate the relationship between self-reported empathy towards animals and perception of animal pain in veterinarians. The authors found a positive association between empathy towards animals and pain scoring in cattle. Similar findings were reported by Ellingsen et al. [32], where empathy was the best predictor of how people rated pain in dogs. In consequence, it is widely accepted that the intensity with which pain is perceived is intricately related to empathy. For this reason, it is not surprising that a number of studies have assessed the empathic responses towards animals through the responses of subjects to images, stories, or videos of animals in diverse situations associated with pain [25,29,96,97]. 

Avoiding negative mental states, such as pain, is among the most important aspects of animal welfare [105]. In consequence, people’s ability to recognize animal pain plays a key role in the assessment and subsequent decision taking for its alleviation in animals [92]. Research about human perception of animal pain and attitudes towards pain in non-human animals has been extensively addressed in populations of veterinarians, veterinary students [29,81,82,106,107,108,109,110], farm animal owners and stockpeople [28,111,112,113,114]. However, to date, there is still a lack of research addressing the association between human perception of animal pain and the welfare state of animals. In relation to the connection between pain perception and animal welfare, Kielland et al. [25] have shown that dairy farmers’ assessments of animal pain, as an indicator of empathy, is a good predictor of animal welfare outcome at the farm level. Knowledge about how working equids owners perceive pain in their equines, and how pain perception may influence the welfare of these animals is limited. However, recently, a study examined the relationship between the welfare state of working horses and their owner’s pain perception. The owner perception of equine pain was assessed through a pain scale, which consisted of photographs that showed equines in diverse painful conditions, and that was previously validated with equine practitioners [35]. The authors found that owners have a high perception of the degree of pain felt by horses and that owners who had a greater perception of pain towards equines had their horses in a better welfare state. In addition, owners’ perceptions of pain towards equines were found to be highly associated with their empathic skills [35]. Therefore, the increased ability of owners to identify painful conditions in their animals can be an important factor determining the welfare status of animals, and thus may affect the way in which equines are treated and cared for. However, further studies in order to conclude a causal association between the owner´s perception of pain in equines and the welfare status of their animals are necessary considering other geo-cultural contexts. 

### 3.3. Attitudes towards Animals and Equine Welfare 

Attitudes are defined as a psychological tendency that is expressed through the evaluation of a particular entity with a certain degree of positivism or negativism [115]. According to Hemsworth and Coleman [116], this definition gives rise to three concepts: first, that those attitudes are directed towards an individual or an object; second, that an attitude is a tendency or disposition towards; and third, that they can be expressed as a positive or a negative appraisal. The authors also remark that attitudes cannot be directly assessed, but need to be inferred from what the individual does or says. Attitudes towards non-human animals has been the psychological construct that has received most attention in the studies of HAIs [77,80,91] and animal welfare [26,31,117,118]. One reason for this is that the attitudes towards animals are not only related to attitudes towards other humans, but are also regarded as an important factor in the prediction of behaviour towards animals [119,120,121], which can, in turn, affect animal welfare and productivity [25,26]. In this sense, numerous studies have shown a sequential relationship between the attitudes and behaviours of people who work with animals in livestock industries and the behaviour, welfare and performance of those animals [43,45,122,123]. Positive attitudes towards animals are associated with a positive handling of them [30,43,122]. This, in turn, may affect the behaviour and welfare of animals through a reduction of the stress and fear response towards humans [21,43,122,124]. 

Scientific interest in human–equine interactions is becoming increasingly popular [60,61,62,63,64,65], and it has been suggested that the welfare of equines may depend on people’s attitudes towards animals [34,125]. To date, there are still only a limited number of studies that have addressed the relationship between human attitudes and equine welfare [34,120,126,127]. In addition, the limited amount of research available in this field have been focused primarily in horses kept for recreational and sports purposes [68,120,126,127], whilst working equids have not received attention. In regard to a potential link between attitudes and equine welfare, some studies have shown that people’s attitude towards animals can affect the behavioural reactions of horses [127] and even physiological parameters, such as heart rate [68]. In a pilot study of veterinary students, Chamove et al. [127] investigated the effect of attitudes towards animals on the behaviour of one single horse. The authors found that positive attitudes towards animals, assessed through of a self-reported scale, were associated with changes in the horse’s behaviour, such as movement and ears position. Specifically, the horse exhibited less ear movement and adopted a forward ear position more frequently, which is indicative of interest or pleasure [127]. On the other hand, a study by Hama et al. [68] found that horses’ heart rates were less reactive when horses were stroked by people who were both confident with horses and exhibited positive attitudes towards animals. This was in contrast with those people who had negative attitudes towards animals, which were capable of inducing an increase in heart rate of the horse. 

Attitudes could be related to how owners perceive their horses, since perception is the mental image that an individual develops of another individual or object, and it depends on prior experiences and learning, becoming the result of a process of selection and interpretation of ideas. Perception is hence the essence of the moral judgements that we perform [128]. In relation to human perception about horses, some studies have suggested that the way in which owners or caretakers perceive their animals (e.g., companion, work or livestock purposes, capable of experiencing emotions) may influence their attitudes towards horses. These, in turn, would affect how they manage and treat their equines, with important consequences on their welfare [17,34,129]. 

It has been pointed out that human-horse relationships are somewhat conflicting because of the dual roles that horses often play [130]. Unlike other species, horses are often considered somewhere between livestock and companion animals, which can generate the acquisition of an instrumental perception (economic or utilitarian perception, i.e., as a work tool) and/or affective (emotional) perception of the animal [17,34,131]. In relation to owners’ perception about their working horses, a recent study undertaken in two regions of Chile found two differing perceptions about working horses: a predominantly affective perception (i.e., horses seen as either member of the family or as a friend) and an instrumental perception (i.e., horses perceived as a working tool) [17]. The results showed that despite the fact that the instrumental perception was more common in one region, the affective perception was predominant and widely shared by both owner populations, suggesting that both perceptions (affective and instrumental) of these horses can coexist. In addition, the study found that most Chilean working horses had a good welfare state, characterized by a low prevalence of both health problems and negative behaviour responses [17]. Based on these results, it could be argued that perceptions that are predominantly affective may generate more positive attitudes towards horses, which can positively affect their welfare. However, the studies in this field are still limited, and further research is required in order to understand the potential relationship between owners´ perceptions, their attitudes and the welfare of their animals.

### 3.4. Locus of Control 

Locus of control, also known as perceived control, is a concept that was developed by Julian Rotter in 1954 [132]. This concept refers to the degree to which people believe that they have control over the outcomes or events in their lives [132]. Rotter [132] proposed that people can be categorized along a continuum from internal to external control. He argued that people with a strong internal locus of control orientation perceive that the outcomes or events in their life are determined by their own personal abilities, effort, and actions, while an individual who possesses a predominantly external locus of control traits believes that the outcomes or consequences of their life are controlled by external forces such as luck, chance, fate or powerful others [132,133].

To date, only one study [36] has investigated the link between owners´ perceived locus of control and the welfare state of working equids. Brizgys [36] sought to identify whether the equid owners’ locus of control was a variable contributing to the physiological and psychological welfare of working equids in Central America. The author found that working equid owners who exhibit more external loci of control traits (i.e., those who perceived that their actions or beliefs depend on their luck, chance or fate) had equids that display a lower behavioural welfare score. However, no significant relationship between the owner’s locus of control and total equid welfare score was found. Brizgys [36] concluded that the implementation of surveys designed to assess the owner´s psychological attributes, such as locus of control, can be useful to implement in educational strategies that seek, in a personalized way, to improve the welfare of working equids considering the particular characteristics of each community.

#### Final Remarks

In recent years, the welfare of working equids has become an increasingly important topic, which is reflected in the high number of investigations worldwide, some of them on a large scale. Additionally, the World Organization for Animal Health (OIE) recently decided to develop the first welfare standards for working equids [18]. These mainly consider the assessment of health state and equids’ behaviour, as well as the management practices and environmental conditions in which these animals should be maintained to ensure a good welfare state. However, neither the standards nor the majority of scientific studies published to date have incorporated the identification and assessment of the human psychological attributes that modulate owner–equine interactions and the potential implications on the welfare of these animals. The aim of this review was to compile evidence demonstrating the importance of understanding those human psychological attributes involved in HAIs and their potential effects on equids’ welfare. 

Studies regarding the relationship between owners’ psychological attributes and the welfare status of working equids are limited. Nevertheless, the few investigations reported in this review indicate that positive effects of high levels of empathy and the greater sensibility to perceive pain on the welfare of these animals exist, demonstrating a clear need for further research in this area. The studies reported in working equids suggest that the owner´s empathy towards animals affects the way in which animals are treated and cared for, with important consequences for their welfare. Thus, empathy assessments and the implementation of empathy education programs should be considered by local governments and non-governmental organizations (NGOs) as an integral component of the strategies that seek to improve the welfare of working equids. This, considering the positive effects that the implementation of these programs has shown in some communities that use working equids [101,102]. 

While research on human-animal interactions began some decades ago, these have intensified in recent years with a considerable number of studies demonstrating that attitudes which stockpeople hold about animals will strongly influence their behaviour towards animals. The attitudes and consequent behaviour of stockpeople affect the animal’s fear of humans which, in turn, has an important effect on the animal´s welfare and performance in livestock industries [19,24]. However, despite the lack of research investigating a link between owners´ attitudes and the welfare status of their working equids, it is reasonable to assume that relationships similar to those reported for the livestock industry may exist. This, as Hemsworth et al. [34] indicate, might imply that owners’ attitudes towards their equines may influence their intentions and behaviours, which in turn may impact upon the welfare of the equines. 

The indicators commonly used in protocols that assess equine welfare include health and behavioural aspects, as well as human management of animals. There is, however, a paucity of research dedicated to identifying and assessing the owner’s psychological attributes that underlie their behaviours, and that may consequently affect the welfare of their animals. The way an owner carries out his or her routine animal care tasks may be influenced by psychological attributes, such as empathy, attitudes towards animals or perceptions about their animal. These attributes can contribute to the overall relationship that owners have with their equines and determine the quality of the human-animal relationship, with its consequent effects on animal welfare. On the other hand, since the assessment of animal welfare includes aspects of the human management of animals, it is reasonable to assume that human behaviour is an important component of animal welfare. Therefore, in agreement with Lund et al. [134], it is necessary to include more researchers from the psycho-social sciences in working equids’ welfare studies. Through an interdisciplinary approach, a better understanding of the factors that modulate human behaviour could be achieved, and it could be easier to design and implement welfare strategies that allow improvement of the well-being of both animals and their owners.

## 4. Conclusions

The studies examined in this review highlight the importance of including the identification and assessment of human psychological attributes that modulate human-animal interactions in the study of working equids’ welfare. Given the important role that working equids have on poverty relief, sustainability promotion, and the resilience capacity of millions of families worldwide, it results necessary to make visible the current state of this group of animals and develop tools and strategies that promote a good welfare state. The assessment of psychological attributes, such as empathy, attitudes towards animals, the owner´s capacity to perceive animal pain and locus of control should be included in welfare assessment protocols of working equids as a diagnostic tool that seeks to improve the welfare of working equids and their owners. 

Identifying human attributes that affect the welfare status of working equids may help researchers, local governments and NGOs to understand the underlying causes of poor welfare and, in this way, appropriate educational intervention strategies could be proposed according to different geo-cultural contexts. Finally, with the purpose of improving working equids’ welfare on a global scale, an integral understanding of those human psychological attributes that influence owners’ behaviour and modulate owner-equine interactions must be established.

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
