# Peer review of "Why Should Human-Animal Interactions Be Included in Research of Working Equids’ Welfare?"

_animals, 2019, doi:10.3390/ani9020042_

Round 1

Reviewer 1 Report

I enjoyed reading your article and felt you presented your thoughts on the role of human animal interaction on working equid welfare very thoroughly, although I wasn’t quite sure what to expect from the abstract. Fortunately, the article was so well written as to explain the concepts alluded to in the abstract very completely. The article has a good flow of the topics and takes logical steps to prove the points. I do not have any suggestions for the major content of the article since I felt you covered and supported the topic adequately. However, I struggled with some very minor grammar and sentence structure details. To help you with this, I converted the pdf of the manuscript to Word document and put my edits right into the text. Hopefully, you can see what my suggestions are in the attached document. Mostly it is minor adjustments to comma placements, use of prepositions and minor grammatical errors. I am recommending the article be published with these minor corrections.

Author Response

We have revised the manuscript “Why should human-animal interactions be included in research of working equids welfare?” according to the reviewers’ comments. We appreciate the time and effort that the three reviewers took in order to improve the quality of the review, at the same time we were very happy to read such positive comments on the work done. Overall, we have included all grammatical suggestions. Sometimes, due to elimination of one sentence, as requested by one of the three reviewers, this sentence could no longer be modified as suggested by a second reviewer. We have done all modifications with the track changes function as suggested.

Some other changes include:

Line 108-110. We have added a sentence in order to complete information about the One Welfare concept.

Line 125-128: We have included a specific example of the implications that positive human-animal interactions can have on animals’ productivity.

Line 334 and Line 368. We have included a comment on interventions for adults in order to improve empathic skills. Most literature related with increasing empathic skills in adults and young adults have focused in people with autism and Asperger, making it difficult to find literature to cite. But this should be something to further study and implement when working with communities in order to obtain data.

Line 462. We have included example of instrumental perception in order to clarify the term

Reviewer 2 Report

This is a much needed review of HAI including that of human-horse interactions.  The review was relatively well organised but a clearer outline of content could have been made more explicitly a the beginning of the introduction. Some of the text was a little repetitive in places but this can be easily addressed. 

An extensive range of contemporary references were sourced and used effectively. 

I have made comments to assist - mostly with improving the english and structure within sections.  I hope that these (n=110) are helpful with the next iteration of this review paper. (File attached).

No statistics needed to be included. 

File attached.

Author Response

(The authors gave the same response as above.)

Reviewer 3 Report

  This is not a research based paper, but instead a review of the relationship between equid welfare and the owners’ empathy for the animals. Most of my comments are minor language ones.

 The author might add a paragraph on how to increase empathy in the adult owners; the only data they report is in school children. Perhaps the equid charities should have the owners listen to a lecture on animal pain etc. in exchange for treatment of their animals. How can we change their “locus of control”?

Locus of control is a new term to this reviewer. It is defined on page 8 but might be defined earlier 

 Define instrumental perception

27 result essential  - do you mean are essential 

Packed work dose that mean carrying loads? Use as pack animals might be clearer 

58 as are their equines 

63 for successful performance

78 improving

85 are you contrasting the livestock industry with the horse and donkey industry

105 has HAI been defined previously in the ms.

127 fear of

133 “given lights’ should be ‘have shed light on ‘

138 used for recreational purposes

141 omit of Despite these

186 don’t capitalize oxytocin

195 empathize 

207 to empathy

220 the associated with 

221 on a human –animal 

222dairy cattle

232 Thy decreased 

233 another similar

236 implications for 

238 relate to

279 ability not sensibility 

“Therefore, the increased sensibility of owners  to identify painful conditions in their animals, moreover, to being related to empathy, can be an important factor determining the welfare status of animals, and thus may affect the way in which equines are treated and cared for.” 
re-write this confusing sentence

290 can be expressed as 

302 omit although 

302 in human-equine

307 With respect

318 of inducing 

324 capable of experiencing

327 because of the dual roles

330 owners’ perception of 

334 despite the fact that that the

337 found that most

360 in a personalized…. of the welfare

369 incorporated

385 attitudes which stockmen hold

403 in agreement with Lund 

407 omit one 

574 do not capitalize within title of article.

Author Response

(The authors gave the same response as above.)
